



ANATOMICAL STRUCTURE OVERRIDE**S** TEMPERATURE CONTROLS ON
MAGNESIUM UPTAKE - CALCIFICATION IN THE ARCTIC/SUBARCTIC
CORALLINE ALGAE *LEPTOPHYTUM LAEVE* AND *KVALEYA EPILAEVE*
(RHODOPHYTA; CORALLINALES)
Merinda C. Nash
Walter Adey
Department of Botany, National Museum of Natural History, Smithsonian Institution,
Washington, DC, USA, 20560
Author for correspondence: nashm@si.edu
Running title: Magnesium and anatomy in coralline algae
Key words: Coralline algae, calcification, biomineralization, magnesium, temperature,
proxy



Abstract
Calcified coralline red algae are ecologically key organisms in photic benthic
environments. In recent decades they have become important climate proxies, especially
in the Arctic and Subarctic. It has been widely accepted that Magnesium content in
coralline tissues is directly a function of ambient temperature, and this is a primary basis
for their value as a climate archive. In this paper we show for two genera of
Arctic/Subarctic corallines, *Leptophytum laeve* and *Kvaleya epilaeve,* that previously
unrecognized complex tissue and cell wall anatomy bears a variety of basal signatures for
Mg content, with the accepted temperature relationship being secondary.  The
interfilament carbonate has lower Mg than adjacent cell walls and the hypothallial cell
walls have the highest Mg content. The internal structure of the hypothallial cell walls
can differ substantially from the perithallial radial cell wall structure. Using high-
magnification Scanning Electron Microscopy and etching we expose the nm-scale
structures within the cell walls and interfilament. Fibrils concentrate at the internal and
external edges of the cell walls. Fibrils ~10 nm thick appear to thread through the radial
Mg-calcite grains and form concentric bands within the cell wall. This banding may
control Mg distribution within the cell.  Similar fibril banding is present in the
hypothallial cell walls but not the interfilament. Climate archiving with corallines can
achieve greater precision with recognition of these parameters. This paper is part of a
series of investigations on controls on Mg uptake and distribution within the crusts of a
range of coralline genera.



## Introduction

Understanding tissue complexity and the structural organization of cell wall calcification
in coralline algae is important for many reasons, including the growing use of these
organisms as climate proxies and concern for the ecological effects of ocean acidification.
There is a burgeoning interest in using coralline crusts as environmental proxies for late
Holocene temperature (Hetzinger et al. 2009, Gamboa et al. 2010, Halfar et al. 2010),
arctic ice sheet coverage (Halfer et al. 2013) and pH changes with time (Krayesky-Self et
al. 2016). Typically magnesium content is used as a key indicator of late Holocene
temperature fluctuations (Adey et al. 2013). Yet despite this utilization of coralline
carbonate crusts for proxy climate research, there has been little study of tissue and
cellular-scale physiology as it relates to the distribution of magnesium within the crust.
Nor are the basic mechanisms of calcification fully understood (Adey 1998). This is in
stark contrast to the status of other calcifiers used for proxy work, e.g. corals (Barnes and
Lough 1993), foraminifera (Bentov and Erez 2005) and bivalves (Wanamaker et al. 2008).
However, these well-known climate proxies have little application in the Arctic Region
of greatest climate change affects (Adey et al. 2013), and without a greater understanding
of coralline calcification physiology, precision proxy analysis of temperature and other
environmental conditions, using coralline algae, is limited.

One of the key roles of corallines is the building of carbonate substrate that underpins
many ecosystems globally.  For example, the thick bioherms found in coral reef
structures (Adey 1978a, b, 1998), the extensive rhodolith beds off South American
(Amado-Filho et al. 2012, Bahia et al. 2010) and Australian (Harvey et al. 2016) shores,



maerl substrate in the Mediterranean (Martin et al. 2014) and the dominant rocky benthos
biostromes and rhodoliths in many Arctic and Subarctic environments (Adey et al. 2013).
There are concerns that as atmospheric $p\mathrm{CO_2}$ increases and consequent ocean
acidification increases, there will be negative impacts on the capacity of corallines to
continue building these important substrates (e.g. McCoy and Kamenos 2014). The pace
of research on the effects of temperature and climate change on coralline algae has
outpaced both the published data on anatomy and our understanding of the biochemical
processes controlling their carbonate skeletal building. For developing reliable past
climate proxy information using corallines and anticipating future climate change impacts
on these keystone calcifiers, as with any other organism, it is first necessary to understand
how these algae organize their tissues, build their skeleton and  control cellular-scale
magnesium content.

While numerous studies of coralline growth rates under a wide range of temperature and
light conditions have been published (Adey and McKibben 1970, Adey 1970, 1973, Adey
and Vassar 1975), little attempt has been made to relate this information to calcification
processes. Also, it is only recently, with the use of higher magnification scanning electron
microscopy (SEM) (Adey et al. 2005, 2015) that the earlier implications of anatomical
complexity (Adey 1964, 1965,1966a) have been fully appreciated. It has been proposed
that calcification is a result of locally elevated pH during photosynthesis leading to super-
saturation and associated mineral precipitation (Ries 2010). However, some parasitic
corallines lack photosynthetic pigments, and have haustoria to derive nutrition from their
hosts, yet present typical tissue and calcified wall structures (Adey and Sperapani 1971,



Adey et al.1974). Also, anatomical and magnesium content studies of Arctic corallines
demonstrate that growth continues in Arctic winter darkness (Halfar et al. 2011, Adey et
al. 2013). There has been an experiment recording continued calcification at night and in
the dark (experiment in progress) indicating that calcification is not likely a straight
forward association with micro-saturation state, as seen in some algae (e.g., *Halimeda*,
Adey 1998, Sinutok et al. 2012).

Following on from the classical coralline studies, maturing around the turn of the 19[th]
century, Adey (1964, 1965,1966a,b) laid out the basic tissue-structured anatomy of
crustose corallines, adding the epithallium, intercalary meristem and cellular elongation
(while calcified) to the classical model of perithallium and hypothallium. Later, SEM
(Adey et al. 2005, Adey et al.2012) demonstrated greater sub-tissue complexity and
added the calcified cell wall components inner wall (IW) and interfilament (IF). In this
paper, we rename the inner wall the cell wall and retain the terminology interfilament,
noting this is equivalent to the middle lamella in higher plants (Esau 1953); interfilament
has also been referred to as interstitial (Ragazzola et al. 2016). We use the abbreviations
PCW and PIF (perithallial cell wall and perithallial interfilament) and HCW and HIF
(hypothallial cell wall and interfilament) to designate the carbonate wall components. It
should be noted that while the interfilament is a minor component of total calcification in
the species of this paper, it can be a major component in some genera (Adey et al. 2013,
2015a).



In this paper, we show for the first time the cellular-scale and anatomical controls on
magnesium distribution within the carbonate skeletons of two Arctic/Subarctic coralline
species. These are *Leptophytum leave* (Stromfelt) Adey, and the epiphytic (and non-
photosynthetic parasitic) *Kvaleya epilaeve* Adey and Sperapani, from the northern
Labrador Coast.  *L. leave* is photosynthetic and forms expansive, but thin crusts (to one
mm in thickness) generally on shell fragments and pebbles in deeper water (Adey 1966a,
1970). *K. epilaeve* is an epiphytic parasite, lacking in photosynthetic pigment, and
producing hypothallial haustoria that penetrate upper perithallial cells of *L. leave* (Adey
and Sperapani 1971). It is similar in physiology to the North Pacific Subarctic parasite
*Ezo epiyessoense* (Adey et al.1974), which, along with its host *Lithophyllum yessoense*,
lies in a distantly related coralline group. *K. epilaeve* is the only known Arctic genus of
algae (Adey et al. 2008) and is absent or of very limited occurrence in Subarctic waters,
where the host continues to be abundant (Adey and Sperapani 1971). Understanding and
contrasting calcification within these two species, both growing in the same temperature,
light and pH conditions, offers an opportunity to examine the wide variance of Mg
content as a function of skeletal anatomy and metabolic processes.

**Methods**
*Sample collection and site information*
The sample was collected on 22$^{nd}$ July 2013, at the commencement of Arctic summer,
from 16-18 m depth at inner Port Manvers Bay, Labrador. The collection site lies at 56º
57.1' N; 61º 32.8' W., near the northern end of the 50 km long Port Manvers Run, a
north/south passageway inside of S. Aulatsivik Island (Fig. 1A). Sea ice is extensive from


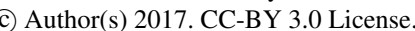


November through early July, and the inter-island passages and bays are covered with
snow-covered land fast sea ice through much of that period. At the collection site, the
bottom was a shell/pebble gravel bed primarily of shell fragments and pebbles encrusted
with *L. leave*, *L. foecundum* and *Clathromorphum compactum*; scattered coarse rhodolith
*Lithothamnion glaciale* and *Lithothamnion tophiforme* were also present (Fig. 1B). *K.*
*epilaeve* occurred on *L. leave* and *L. leave* grew on both sides of the shell fragments.
Salinity was measured using electronic induction instrumentation and was 30 ppt.
November to July near surface water temperatures, below the sea ice, are within the -1.5
to -1.8º C range. Bottom summer temperature measured at the site on 22nd July 2013 was
0.5ºC. Since this is relatively early in the summer season, peak temperatures are likely to
be between 3-5ºC (Adey et al. 2015) with a mean growing season temperature of ~ 2 º C.
This mean estimate is based on measurements from eight sites in the region (182 km S to
35 km N) with surface to bottom temperature records for 1964 (Adey 1966c) and 2013
(Adey et al. 2015). These ranged from 1.9 to 5.6º C during summer at 15-20 m. The
snow-covered land fast sea ice overlying the gravel rhodolith bed from which the samples
were taken likely precludes significant solar energy from reaching the bottom for eight
months of each year.

Species identification was made by WHA.   The original sample is 2013-11(1) at the
National Museum of Natural History.

**Analytical methods**
*Scanning electron microscopy- energy dispersive spectroscopy (SEM-EDS)*



The CCA sample was fractured, mounted using carbon tape and platinum coated prior to
scanning electron microscopy energy dispersive spectroscopy (SEM-EDS).  For these
analyses, we used a Zeiss UltraPlus field emission scanning electron microscope
(FESEM) equipped with an HKL electron backscatter diffraction (EBSD) operated at 15
kV, 11 mm working distance.  SEM was carried out at the Australian National University
Centre for Advanced Microscopy. SEM-EDS was used for spot analyses to quantify the
elemental composition of representative parts of the CCA crust. A range of SEM settings
were used for imaging.  The more common secondary (SE) electron showing topography,
backscatter electron imaging  (BSE) which shows higher magnesium areas as darker
carbonate and is useful for rapid visual identification of mineral distribution.

A second round of EDS was undertaken using a NOVA NanoSEM FEI at the National
Museum of Natural History's Department of Mineralogy. Typically EDS measurements
are made using 15 kV (Nash et al. 2011) so that there is enough energy to dislodge
electrons from a range of elements, e.g. from lighter magnesium up to heavier strontium.
The EDS beam interacts with a roughly spherical-shaped region of carbonate beneath the
surface. This region is referred to as the interaction volume. At 15 kV the interaction
volume is ~ 3 μm in diameter whereas the average cell wall thickness ranges from only
500 nm up to ~2 μm (occasionally thicker, up to 3 μm).  Interfilament in these species
may be only a few grains wide, 200-500 nm up to 2 μm.  These narrow areas of interest
in contrast to the larger beam interaction volume, pose a problem for obtaining accurate
Mg measurements for only cell wall or interfilament. For example, a measurement of the
cell wall may include minor amounts of carbonate from the adjacent interfilament and



vice versa. Generally even with this beam crossover, in our experience 15 kV is sufficient
to identify a significant offset in magnesium while still collecting information that may
be of interest such as strontium levels.  However, where there are only a few grains of
interfilament, as in the *L. leave,* the 3 μm interaction volume is problematic. A range of
EDS settings were tested aiming to reduce the beam interaction volume so that Mg
content for each the cell wall and the interfilament could be individually measured
without the beam crossing into the adjacent substrate. A setting of 7 kV, working distance
6.4 mm and 1 nA current was used to measure the interfilament grains in the *l. leave* with
a count time of 20 seconds. The sample was carbon coated. This was calculated to have
an interaction volume of  <1 μm. These results are reported separately to the main data
set.

*Sample preparation*
Initially the crust was fractured using shears and mounted in superglue.  After first
imaging of the fractured crust, the sample was polished using 2000 gsm wet and dry
sandpaper then sonic cleaned in unbuffered deionized water for 2 minutes.  This
preparation was used for SEM EDS measurements; 8-9 measurements were made for
each carbonate type of interest. Subsequently the sample was sonic cleaned in unbuffered
deionized water for 20 minutes.  The deionized water has a pH of ~6.5.  When cleaned
for 2 minutes the surface is very lightly etched allowing differentiation between different
Mg-calcite morphologies without altering the measured Mg content.  After cleaning for
20 minutes there is a visible difference in the surface with much of the interfilament Mg-
calcite and smaller grains removed allowing imaging of nm scale cellular structures.




X-ray diffraction methods
Powder XRD was carried out using a SIEMENS D501 Bragg-Brentano diffractometer
equipped with a graphite monochromator and scintillation detector, using CuKα radiation.
A subsample was broken off the edge of the crust. This piece included *L. leave* with
surficial *K. epilaeve*. The sample was ground using a mortar and pestle. Fluorite was
added as an internal standard. The sample was not bleached and acetone was not added
during the grinding as this has been found to occasionally induce alteration and
precipitation of other minerals in other coralline samples we have worked with. Scan
interpretation for mol% $MgCO_3$ followed the methods described by Nash et al. (2013).

*Temperature calibration*
Data for the graph in figure 5 taken from Halfar et al. (2010, 2013).

**Results**
*SEM imaging overview*
The specimen of *L. laeve* encased an aragonite carbonate shell. (Fig. 2A).  The crust is
approximately 500 microns thick (Fig. 2B) with a basal hypothallus ~80 microns thick. *K.*
*epilaeve* has been considered to be an adelphoparasite, a species very closely related to its
host. Although diminutive, and superficially appearing as scattered white sand grains, *K.*
*epilaeve* can densely coat *L. leave*.  Although often appearing as densely crowded
conceptacles, it can possess the full basic array of anatomical features: hypothallium,
perithallium and epithallium (the latter mostly absent, Adey and Sperapani 1971) (Fig.
2B). *L. laeve* typically has an epithallium that is one cell layer of rounded ovoid, thin



walled cells that are often absent in SEM sections.  The *K. epilaeve* grows directly on the
*L. laeve* meristem (Fig. 2C, D) and there was no evidence of excavation required (by
borers or grazers), prior to settlement. This suggests that unlike the typical sloughing
relationship with epiphytes wherein epithallium builds up under the epiphyte until it
sloughs off, the *L. leave* does not recognize *K. epilaeve* as foreign. The perithallial cell
walls of *L. laeve* contain radially-oriented grains of Mg-calcite; the interfilament is thin
and has carbonate grains randomly orientated in a plane parallel to the filament axis or
cell top/ bottom. The interfilament shows up strongly as stripes on vertical fracture
sections (Figs. 2B, C). Note for easiest viewing of the fine structures, the figure images
are best viewed on screen rather than in print.

The first layer formed by the *K. epilaeve* has angular grains parallel to the *L. laeve*
surface (Fig. 2E). The bottom part of the cell wall is without radial structure and has
submicron beads appearing to calcify along and within organic fibrils (Fig. 2E).  Organic
fibrils are visible between the basal layer of *K. epilaeve* carbonate grains and the
meristem of the *L. leave* (Fig. 2F) suggesting a method of attachment in addition to the
haustoria developed by some hypothallial cells (Adey and Sperapani 1971). There were
no haustoria visible in our SEM sample. Fine radial grains typically observed in cells of *L.*
*leave* beneath the meristem were not apparent in the cell walls of the *L. laeve* meristem
(Fig. 2E,F) suggesting this surficial carbonate may have been altered or remineralised
during the attachment process.

*SEM-EDS*



Measurements for magnesium content in *Leptophytum leave* were undertaken on both the
upper (side with conceptacles) and under (without conceptacles) crusts (Fig. 2A, D). The
parasite, *Kvaleya epilaeve* was present on both surfaces (Fig. 2A, Fig. 3A, B).
Measurements of *K. epilaeve* were made on the underside.

The Mg content of the perithallial and hypothallial cell walls of *L. leave* was measured
(Fig. 2 A-D) as well as what appeared to be a transitional cell type between the basal
hypothallus and the typical perithallial cells (Fig. 2 D-F). These transitional cells are
within the perithallus but have thin cell walls similar to the hypothallial cells.  There are
clear visual differences between the cell walls of the three cell types.  The perithallial cell
walls are 1-2 microns wide with clearly radial Mg-calcite (Fig. 2B, F).  The basal
hypothallial cells are elongated relative to the perithallial cells and their cell walls are
narrower and do not always show radial cell wall structure (Fig. 2C). The transitional
cells have elongate cells relative to the perithallus but less so than the hypothallus, and
their cell walls are thinner, $\sim 0.5 - 1$ micron and do not show radial structures. The
interfilament of *L. laeve* has only a single layer of Mg-calcite grains (Fig. 2B, F), as noted
above showing as a thin line on longitudinal axial fractures; fractures along the
interfilament appear as conspicuous vertical stripes (Figs. 2C).

The *K. epilaeve* in the portion of the sample mounted for SEM did not present the typical
elongated hypothallial cells as shown by Adey and Sperapani (1971), as this cut is not
longitudinally placed on a growing lobe. The key difference between the perithallus of
the *L. laeve* and *K. epilaeve* was the presence of wide (1-2 microns) areas of interfilament



in the *K. epilaeve* (Fig. 3A, F, 4A, B). In many corallines (Adey et al. 2005), including
the *L. laeve* studied for this paper there is only a single layer of interfilament grains, and
these present as vertical stripes on vertical fractures (Fig. 2B). EDS measurements were
taken for both the *K. epilaeve* cell wall and interfilament (Fig. 3A, B).  As the interaction
volume of the EDS beam is ~ 3 microns (Methods) and the cell wall and interfilament
thickness range from 1-3 microns, the values measured for both may include small
amounts of the other, although every effort was made to place the beam on the widest
part of the appropriate band.  A second set of measurements was taken for the *L. leave*
cell wall and interfilament using lower kV and the results are reported separately.

*Mg content*
Bulk whole sample content of Mg, determined by powder XRD was 10.8 mol% $MgCO_3$
(Mg/Ca 0.13). This XRD Mg content is within the range for average winter and summer
Mg contents for *Clathromorphum compactum* collected from Arctic Bay, Kingitok and
Quirpon (Halfar et al. 2011, 2013).  The EDS-determined average Mg content ranged
from 9.1 (*K. epilaeve* Perithallial interfilament) to 16.7 mol% $MgCO_3$ (*L. leave* upper
Hypothallial cell wall), (Table 1, Fig.6). The highest measured individual Mg content,
19.6 mol% $MgCO_3$, was in the *L. leave* upper crust HCW.  Generally the Mg content of
interfilament was lower than cell walls, and perithallial cell walls had the highest Mg
content. The lowest values were for the *K. epilaeve* PIF and PCW, 9.1 and 10.1 mol%
$MgCO_3$ respectively, not significantly different at significance level of 0.05 but are
significantly different at significance level of 0.1 (p= 0.068) (Table 2). Keeping in mind
the values for the cell wall and interfilament include a small amount of carbonate from



the other, we consider the p=0.068 result likely does represent a true significant
difference between the two. The PCW for the *L. laeve* was slightly higher at 11.2 and
12.9 mol% $MgCO_3$ (under and upper crust respectively), these were not significantly
different from each other (p=0.112).  The combined average of the upper and under *L.*
*leave* cell walls (12.2 mol% $MgCO_3$) was significantly higher (p=0.025) than the *K.*
*epilaeve* cell wall.  However, comparing only the *L. leave* cell wall of the under crust, the
same side as the *K. epilaeve,* there was no significant difference (p=0.124). The greatest
difference between the upper and under *L. laeve* crust was found between the hypothallial
cell walls.  The under HCW averaged 12.3 mol% $MgCO_3$, whereas the upper HCW was
4.4 mol% higher at 16.7 mol% $MgCO_3$. The upper HCW was significantly higher than
the *L. leave* PCW's but not different from the transitional CW's (15.6 mol% $MgCO_3$).
Based on the graph in figure 5 this upper range of Mg would equate to temperatures
above 9.3°C, more than double the known summertime highs at the sampling site.

The results for comparison of the cell wall and interfilament grains in the *L. leave* using 7
kV showed the interfilament, 8.5 mol% $MgCO_3$ (n=6), was significantly lower (p=0.001)
than the cell wall, 11.1 mol% $MgCO_3$ (n=8).

**Structural features**
*Cell wall*
Within the radial Mg-calcite structure (PCW) of the *K. epilaeve*, a concentric banding
pattern is present (Fig. 7 A-C). The radial Mg-calcite grains are not always one
continuous long grain. The banding is aligned to the presence of organic fibrils that



appear regularly throughout the PCW (Fig. 7B).  Organic fibrils, ~10 nm thick, are
parallel to the cell wall edges.  These are spaced 30-40 nm apart throughout the middle of
the cell wall.  It appears that the fibrils are mineralized. At the outer edges of the cell wall
the number of fibrils increases and appear as a dense mesh approaching a membrane (Fig.
7B, C) that is infilled with carbonate. The parallel fibrils are connected to the radial Mg-
calcite grains, appearing as if to continue through the grain (Fig. 7C), similar to fence
wire threading through fence posts at pre-defined spacing. There are also fibrils that
drape over the grains. Where the fibrils concentrate to a mesh, this is also calcified but
with smaller grains without regular shape. In the *K. epilaeve* interfilament (PIF), the
grains are aligned to the cell wall surface (Fig. 7C). Fibrils also run through the PIF and
attach to the interfilament grains but not with the regular pattern seen in the cell wall.
Looking at a cross section of the cell wall from the top down (Fig. 7D), the fibrils can be
seen to form a dense mesh.



Similar features are visible in the *L. laeve* PCW (Fig. 8A, B), although the organic fibrils
are not as well exposed.  Possibly these cell wall grains are less susceptible to dissolution
in the etching treatment making it more difficult to expose the organic features. The
radial cell wall grains appear anchored to the external edge of the cell wall, immediately
adjacent the interfilament.

After etching for 20 minutes, more of the organic fibrils are exposed in the *K. epilaeve*
interfilament (Fig. 9A) revealing a porous membrane. PIF grains have angular edges in
contrast to the rounded sides of the cell wall grains.  The *L. laeve* perithallial
interfilament has rice-grain shaped Mg-calcite flattened against the external side of the
cell wall (Fig. 9B) with attachment fibrils.  Fibrils are visible stretching between the
flattened interfilament grains on adjacent cells (Fig. 9C).

Hypothallial cell walls at 200-500 nm wide are much thinner than perithallial cell walls
(Fig. 10 A-C). The HCW internal structure appears roughly radial (Fig. 10 A- C). But, the
radial structure is not always well developed with parts of the HCW exhibiting a distinct
break down the middle of the radial structures (Fig. 10C).  There are fibrils parallel to the
cell wall appearing to go through the wall grains similarly to the perithallial cell walls.
Interfilament grains are present, as in perithallial cells (Fig. 10B, C).   The HCW wall can
have two clearly defined morphologies (Fig. 10C). The wall adjacent to the interfilament
is narrowest at ~200 nm, has closely spaced organic fibrils and is poorly calcified
compared to the inner part of the wall (300-400 nm wide) and appears more like a
mineralized membrane. The wider inner part of the cell wall has radial grains but without


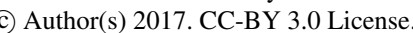


the well-defined shape of the PCW radial grains.  Similar to the perithallial cell walls,
there are fibrils appearing to thread through the hypothallial cell wall grains.

The transitional cells between the hypothallus and perithallus have features from both
types present (Fig. 10D).  The cell walls can be narrow, <200 nm, poorly mineralized
similarly to the outer part of the hypothallial cell wall.  Parts of the cell wall resemble the
perithallial cell walls, with radial grains and wall width of nearly 1 micron, although
along the same wall this changes to ~200 nm wide and a poorly mineralized membrane.
The parallel fibrils are also present within the transitional cell walls. Interfilament grains
are present comparably to those between hypothallial and perithallial cells.

**Discussion**
*Site temperature, ecology and growth*
The site of collection for this specimen (Fig. 1A) is a pavement of coralline encrusted,
roughly flat to ovoid shells and pebbles often with dish shapes. Many, such as the
specimen employed in this study have a concave surface (due to the original mollusk
shape). The benthic surface that we show in figure 1B is likely quite stable with time in
the moderate reversing tidal current environment of the site. The conceptacles of *L. leave*,
requiring considerable solar energy for construction; all appear on the upper side of the
specimen and further assist our determination of orientation. Since the sea ice does not
clear the area until late June or early July, solar energy has already peaked, by the time
the benthos at 15-17 m receives significant light. Effectively, the growing season is July
through November, and with a mean growing season temperature of < 2º C. Based on the





lateral growth rates (5-7 μm/day) found by Adey (1970), a season of lateral growth would
provide less than one mm of extension. As we discuss below, the vertical growth in this
species is slower than the lateral growth. The layering seen in figure 1B likely represents
4-5 years of vertical growth. At 80-100 μm of perithallial addition/year, this relates well
to the 100-200 μm /year found with extensive data in the same region for
*Clathromorphum compactum* (Adey et al. 2015b).

Considering that *Leptophytum leave* crusts can be many cm broad and rarely exceed 500
μm in thickness, except by overgrowing of earlier crusts, it can be assumed that after
initial formation, upwards perithallial growth is either very slow, perhaps limited by the
development of conceptacles for which considerable photosynthate must be dedicated. *L.*
*leave* is a deep water species (Adey 1966a, b, 1968, 1971) and requires little solar energy
to grow and carry out its life cycle; however, as shown by Adey (1970), the rate of
hypothallial extension falls with light reduction, and it would be expected that growth on
the underside of a shell-encased fragment would be present but less than that on the upper
surface.

*Temperature and magnesium*
One of the challenges using samples collected at a single point in time is that the growth
history cannot always be precisely tied to previous points in time and temperature.  As
discussed in the previous section, this crust likely represents 4-5 years of growth. Thus
the XRD mol% $MgCO_3$ is an average for that period. The individual EDS measurement



spots cannot be tied to a particular time of year or temperature.  However, the annual
temperature range is not large, estimated to be ~ 4 ºC across the growing season.

The EDS-determined average Mg content for each carbonate type had a range of 7.6
mol% MgCO$_3$, from 9.1 (*K. epilaeve* interfilament) to 16.7 mol% MgCO$_3$ (*L. laeve* upper
crust hypothallus). The *L. laeve* upper hypothallus has 84% more Mg than the *K. epilaeve*
interfilament.  Although the exact time and temperature of formation for each component
is not known, the temperature range (~4 ºC) alone is highly unlikely to explain the Mg
difference.  Studies on Mg content in CCA for temperature proxies have used regressions
with temperature records to determine a range of responses from 0.266 mol %
(Williamson et al. 2014), ~1.0 (Halfar et al. 2000; Darrenougue et al. 2013) to 1.76 mol%
MgCO$_3$ (Kamenos et al. 2008) per degree celsius of temperature increase.  Only the
Kamenos et al. (2008) calibration is close to explaining the range here.  However, that
calibration was for branches of the rhodolith *Lithothamnion glaciale*.  Using temperature
calibrations for crust CCA in experimental treatments, where temperature was the only
condition changed (Diaz-Pulido et al. 2014; Nash et al. 2016), a calibration of 0.33
mol%/°C is obtained.  This rate is in agreement with results from Williamson et al.
(2014), Chave and Wheeler (1964) and Adey (1965). Using 0.33, a shift of 7.6 mol%
equates to 23°C of change, nearly four times greater than the maximum annual range at
this site. The magnesium offsets in different parts of the crust are clearly aligned to
anatomical features and not controlled by temperature.  Within these offsets there may
still be a response to temperature over the seasons, but it was beyond the capacity of this
study to investigate seasonal changes.  It is noteworthy that the upper crust hypothallus



average of 16.7 mol% $MgCO_3$ is equivalent to new surface crust of tropical *Porolithon*
*onkodes* grown at 30° C (Diaz Pulido et al. 2014)

*Structural features*
There are three main types of calcified structures within the vegetative tissues of
*Leptophytum leave* and *Kvaleya epilaeve*: (1) the radial Mg-calcite within the cell walls
of the perithallium, (2) the interfilament in both the perithallium and hypothallium and
(3) the thin hypothallial cell walls. Each has distinctively different features and
magnesium content. The more elongate (and thinner-walled) cells of the hypothallus have
been reported for other species of Melobesioideae (Adey 1964, 1965, 1966a).  However,
this is the first study to show that the internal cell wall Mg-calcite structure and their
magnesium content differs from perithallial cell wall. Probably these thinner elongated
hypothallial cell walls are a result of relatively rapid growth during lateral extension.
There are numerous examples documenting higher Mg in parts of crusts that have grown
faster during the warmer seasons (e.g. *Clathromorphum compactum* and *C. nereostratum*
by Adey et al. 2013).  In this case there is no elevated temperature. The mechanistic
process by which more Mg is incorporated into the HCW and how this relates to growth
rate is not known.

*Calcification and photosynthesis*
The parasitic epiphyte *K. epilaeve* is not known to photosynthesize. The similarity of cell
wall and interfilament features to those of the photosynthesizing host, *L. leave*, suggests
that the precipitation of the Mg-calcite is not directly driven by photosynthesis as has





been suggested for coralline algae (Ries 2010) and demonstrated for calcifying green
algae *Halimeda*, (e.g. Adey 1998, Sinutok et al. 2012). Rather, considering also the
evidence for continued calcification during the Arctic winter (Halfar et al. 2011, Adey et
al. 2013), it seems likely the first control is the provision of the organic substrate that
subsequently either becomes calcified or induces calcification. This does not negate the
possibility of increased calcification as photosynthetic rates increase (e.g. Borowitzka

449    1981).

*Banding and magnesium uptake*
The concentric banding of organic fibrils within the perithallial cell wall is interesting
from a magnesium perspective.  The dominant visual morphological pattern is the radial
Mg-calcite crystals. In contrast, other work indicates the dominant pattern of Mg
distribution within the cell may be unrelated to the radial features. Concentric zonations
of higher Mg content have been shown, using back scatter electron imaging, in cell walls
of tropical *Porolithon onkodes* (Nash et al. 2011). Ragazzola et al. (2016) using
NanoSIMs, also showed clear concentric banding of Mg within summer cell walls of
*Lithothamnion glaciale*. These published observations together with the results in this
study suggest there could be a strong organic control on Mg distribution within the cell,
with this being related to the concentric fibrils. Possibly the fibril organics enable higher
Mg incorporation than the organics involved in the radial structures.  Ragazzola et al.
(2016) further documented a decreased prominence of Mg banding in winter cells of *L.*
*glaciale* and for those grown in $CO_2$ enriched conditions.  Results from our study offer an
insight as to possible temperature or $CO_2$-driven ultrastructure changes that may result in
decreased Mg content.  If the banded fibrils observed in this study are normally similarly



present in the *L. glaciale,* then an absence of the Mg bands for their winter and elevated
$CO_2$ treatment suggests that these fibrils could either be absent, or the organic structure or
composition has changed and no longer enables elevated Mg.

*Relevance to Climate Archiving*
This study has several implications for climate archiving using corallines. Most
importantly, anatomical controls can override temperature influences on Mg composition.
Thus, any study of CCA for temperature archiving must take into account changes in
anatomy throughout the measured areas.  While hypothallial areas can usually be easily
excluded from most climate archiving (but see Bougeois et al. 2015), less obvious
anatomically different tissues such as the elevated Mg transitional cell walls may not be
noticeable at low magnification. This may lead to a false positive result identifying such a
region as reflecting a time of higher temperature. As well as these tissue-scale differences,
the cellular scale differences may also need to be considered.  Any seasonal change in
relative proportion of CW to IF can shift the [Mg] in absence of any temperature-
influenced change.  For example if CW = 10 mol% $MgCO_3$ and IF = 8 mol% $MgCO_3$,
and crust changes from 90:10 CW:IF to 50:50 this would equate to a change in of 9.8 to 9
mol% for measurements of bulk crust (i.e. spot sizes larger than the cell size, or smaller
spot sizes averaged without reference to their anatomical placement). This change
equates to a 2-3 degrees using a temperature calibration of 0.33 mol% $MgCO_3$ °C. Should
the difference in cell wall and interfilament mol% $MgCO_3$ be larger, then the total
average will change more substantially. Furthermore, the bulk magnesium results for
different CCA species with differing proportions of cell wall:interfilament from the same





temperature environments will have a range of non-temperature related Mg content that is
controlled by the cell wall:interfilament. This change in structure, if seasonally correlated,
will be indirectly related to temperature, but there may be other influences such as light.
Thus, the best CCA temperature climate archives, as compared to seasonal archives, are
likely to be those with the least seasonally varying ultrastructure changes.

Understanding the combined contribution of anatomical and temperature changes to
measured magnesium may help explain the variation of Mg-temperature calibrations in
the published literature. Typically it is the rhodoliths that show the highest response of
Mg to temperature, e.g. *Lithothamnion glaciale* at 1- 1.76 mol% $MgCO_3$ (Halfar et al.
2000; Kamenos et al. 2008) per degree celsius of temperature increase compared to
*Clathromorphum compactum* at 0.7 mol% $MgCO_3$ (Halfar et al. 2010). The *L. glaciale*
has distinct seasonal changes shifting to a clear band of elongated cells during summer, in
contrast, anatomical changes in *C. compactum* (Adey et al. 2013) are not so extreme.

*Suggestions for improving analytical methods*
Our work is ongoing in this area of research and as more species and ultrastructure are
studied we expect to be able to provide more detailed guidance on utilizing Mg from
CCA for climate proxies. However, in the interim, there are several steps that could be
incorporated into routine analyses to improve the accuracy of Mg climate proxies. Firstly,
it should become a routine part of analyses that the ultrastructure is assessed to determine
if the ratio of cell wall to interfilament carbonate changes regularly with seasons. Second,
when possible as well as the larger spot sizes used in sampling transects, e.g. 10-20



microns, make discrete spot analyses using the smallest reliable interaction volume
possible to determine indicative Mg offsets between the cell wall and interfilament so
that this can be adjusted for if necessary, in the final interpretation. Third, ensure that
hypothallial growth is not included in sampling transects. Usually the basal hypothallus is
easily avoided, but secondary hypothallus and transitional cells may be harder to avoid
without careful SEM analysis.

**Conclusion**
It appears that within these CCA, there is a strong control on the uptake of Mg in relation
to the different anatomical components.  This is in contrast to the suggestion by Ries
(2010), based on Mg:Ca in seawater manipulation experiments, that corallines exert little
or no control over their Mg uptake other than to specify the polymorph. It would be
interesting to identify if each of interfilament, perithallial and hypothallial cell walls
reacted similarly to changes in temperature and seawater Mg:Ca, or if there were
differences in anatomical controls. Crucially, it is necessary to keep in mind the
biological controls on Mg uptake when using CCA Mg changes as a climate proxy.

**Acknowledgments**
Thanks to the Centre for Advanced Microscopy at the Australian National University and
the Mineral Sciences department at the Smithsonian Institution for assistance with SEM-
EDS.

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




## Tables

| | *K.epilaeve* | | *L. laeve* | | | | | |
|---|---|---|---|---|---|---|---|---|
| | IF | CW | CW Under | CW Upper | CW comb. | Under Hyp. | Upper transit. | Upper Hyp. |
| **mol% MgCO₃** | 9.1% | 10.1% | 11.2% | 12.9% | 12.2% | 12.3% | 15.6% | 16.7% |
| **St. Dev.** | 1.0% | 1.2% | 1.2% | 2.5% | 2.2% | 0.7% | 1.7% | 1.7% |
| **Mg/Ca** | 0.100 | 0.113 | 0.126 | 0.149 | 0.138 | 0.140 | 0.185 | 0.200 |

**Table 1: SEM-EDS results. Conversion of mol% to Mg/Ca is included.**

| | Average mol% and n | *K. epilaeve* IF | *K. epilaeve* CW | *L. laeve* under CW | *L. laeve* upper CW | *L. leave* CW both | *L. laeve* under Hyp. | *L. laeve* upper Hyp. |
|---|---|---|---|---|---|---|---|---|
| *K. epilaeve* IF | 9.1 % n=9 | | | | | | | |
| *K. epilaeve* CW | 10.1% n=8 | **0.069** | | | | | | |
| *L. laeve* under CW | 11.2% n=8 | | 0.129 | | | | | |
| *L. laeve* upper CW | 12.9% n=9 | | **0.012** | 0.112 | | | | |
| *L. leave CW both* | 12.2% n=17 | | **0.024** | | | | | |
| *L. laeve* under Hyp. | 12.3% n=8 | | | **0.052** | 0.470 | 0.914 | | |
| *L. laeve* upper Hyp. | 16.7% n=8 | | | | | **<0.001** | **<0.001** | |
| *L. laeve* upper trans. | 15.6% n=8 | | | | | **<0.001** | **<0.001** | 0.259 |

**Table 2: T-test *p* values for 15 kV spot EDS.**

## Figures



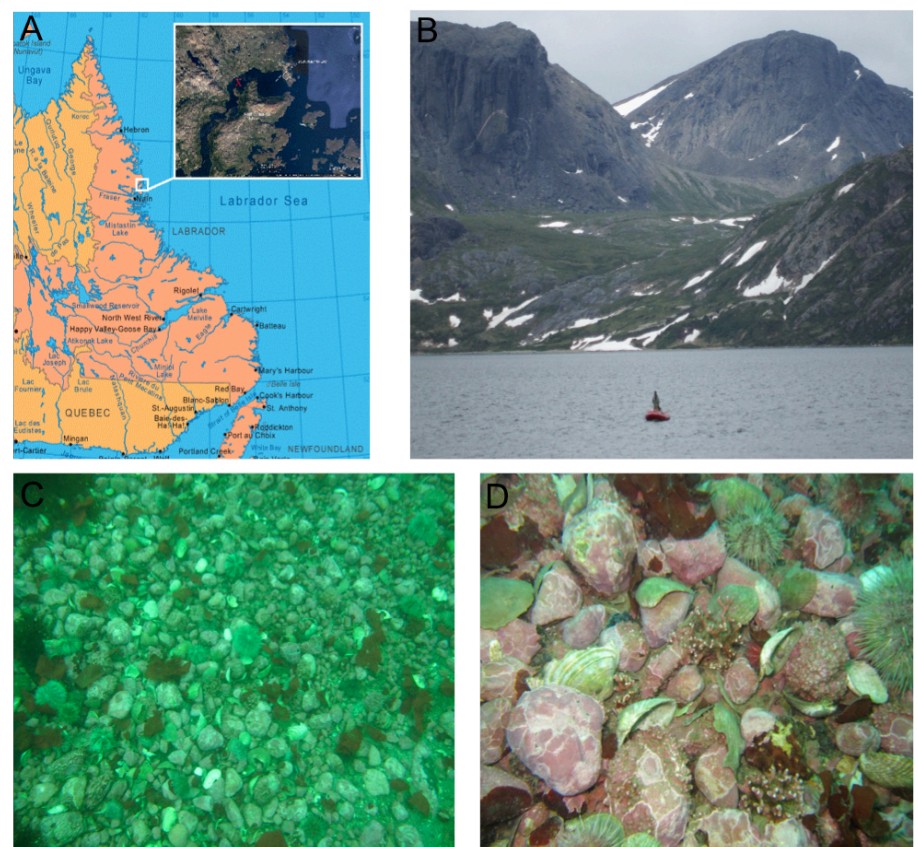


Figure 1: A. Port Manvers Bay Station, Labrador. B. Collecting site in western Port Manvers Bay. C.
Pebble/shell bottom with occasional rhodoliths at 15-17 m.  Coralline covered pebbles range from about 5-
10 cm diameter. D. Close-up of bottom shown in figure 1C.




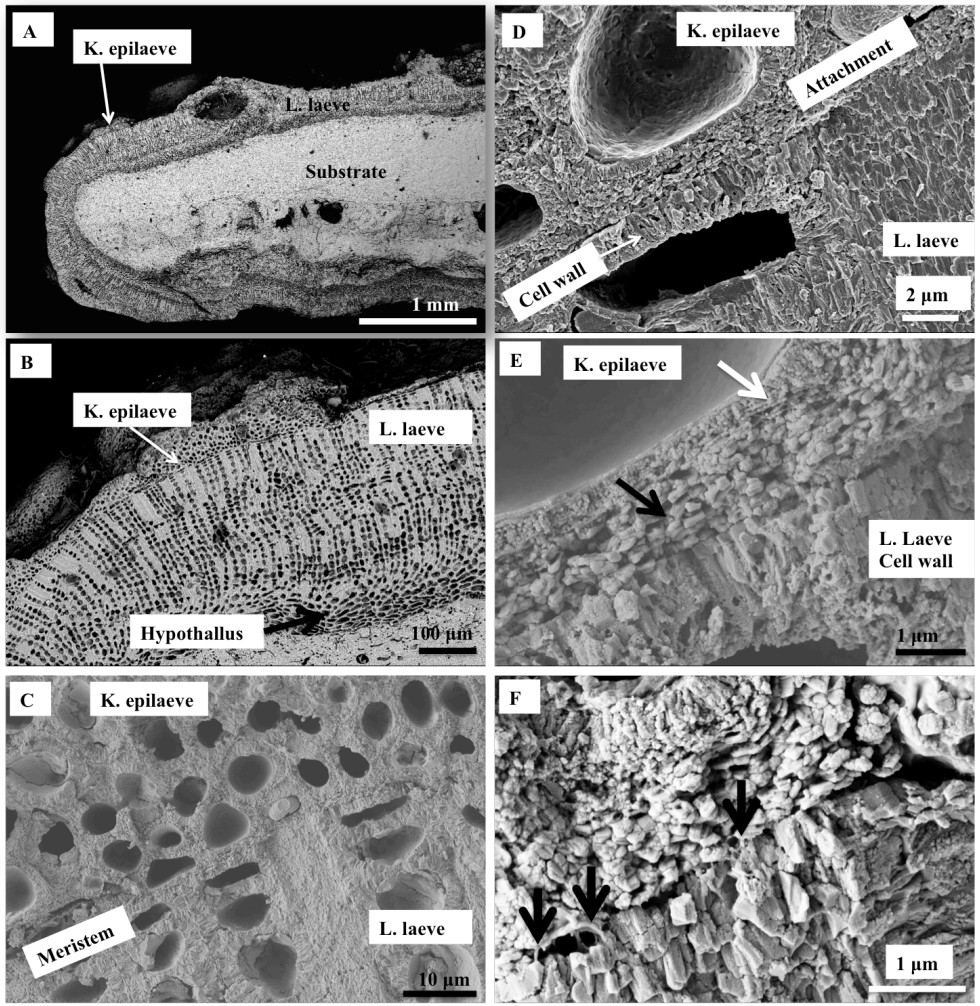


Figure 2: Overview of *K. epilaeve* on *L. laeve*. **A**. Overview (BSE). *L. laeve* has been partly overgrown by
*K. epilaeve*. **B**. Closer up (BSE) *K. epilaeve* has a very thin perithallium with thicker buildup for its
conceptacle. **C**. Close up (SE) and **D** showing attachment zone of *K. epilaeve* hypothallus on the meristem
of the *L. laeve*. **E**. (SE) The cell wall in the *L. laeve* is roughly radial whereas the *K. epilaeve* cell wall does
not appear properly mineralized with nm-scale beads of Mg-calcite along what appears to be organic fibrils
(white arrow). The *K. epilaeve* Mg-calcite layer at the attachment zone has coarse angular grains roughly
parallel to the *L. laeve* surface (black arrow). **F**. (SE) Organic fibrils are visible (black arrows) between the
base of the *K. epilaeve* and the surface of the *L. laeve* suggesting this is the attachment mechanism.





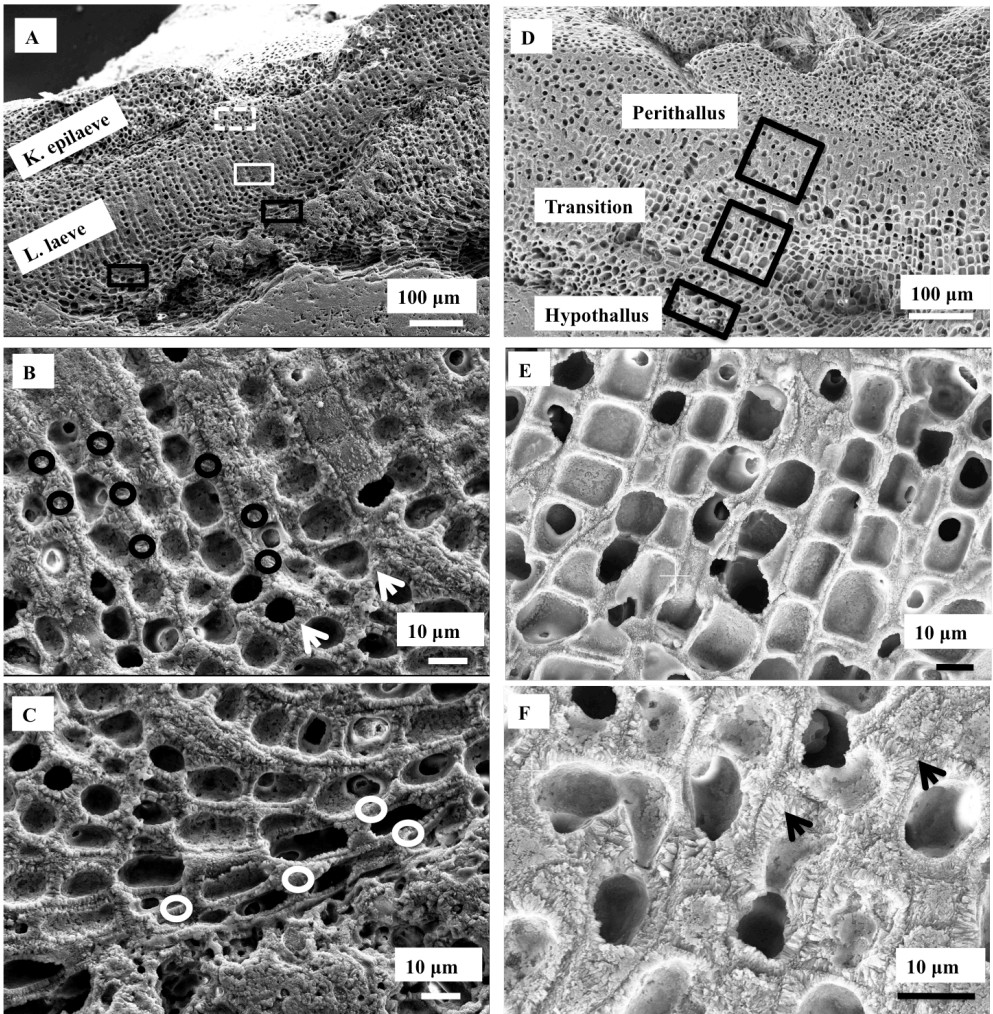


Figure 3: Overview of *L. laeve* and *K. epilaeve* and  EDS sites in *L. laeve*. **A-C**. Sites on the underside of

the pebble. **D-F**. Sites on the upper side of the pebble. **A**. White dashed box- cell wall and interfilament in

*K. epilaeve*. White box– perithallial cell wall *L. laeve*. Black box- hypothallus *L. laeve*. **B.** EDS sites for

cell wall measurements of *L. laeve*. Circle size indicates approximate area of measurement (3 microns).

Cell wall radial Mg-calcite (arrowheads). **C**. EDS sites for hypothallus (right box in A). **D**. EDS sites on

sample upper side for *L. laeve*. E. *L. leave*. **F**. *L. leave.* Cell walls in upper side are visually comparable to

cell walls in underside with radial Mg-calcite (arrowheads) in cell walls and minimal interfilament.





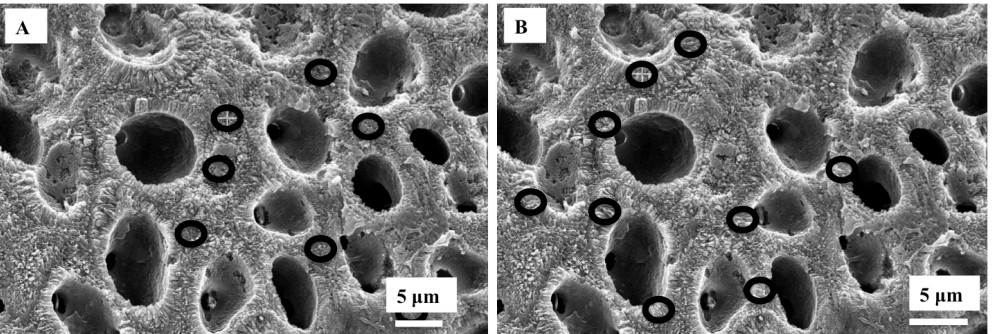


Figure 4: Detail of EDS in *K. epilaeve* (dashed white box in Fig. 2A) **A**. EDS sites for interfilament. **B**.
EDS sites for cell wall.

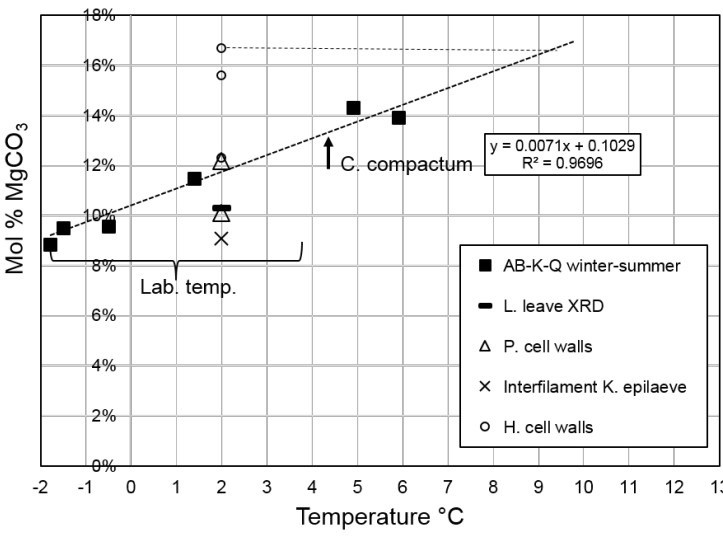


Figure 5: *L. Leave* and *K. epileave* Mg content relative to *Clathromophum compactum* from Arctic Bay,
Kingitok and Quirpon (Halfar et al. 2010, 2013). Lab – Labrador sea.  Heavy dashed line- best fit for *C.*
*compactum*. Light dashed line- indicates the temperature equivalent on the *C. compactum* line for the L.
leave hypothallial Mg-content.





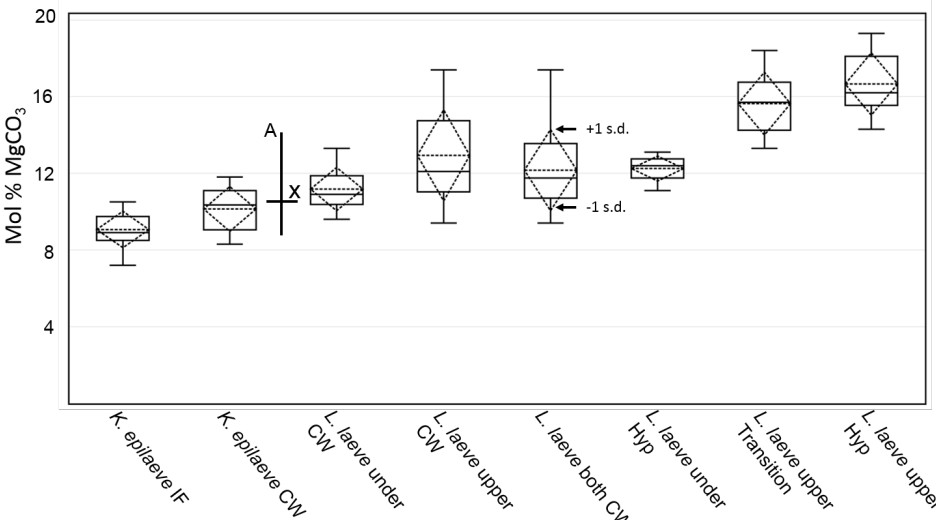


Figure 6: Box plot of EDS mol% $MgCO_3$ results. Box represents the 2nd and 3rd quartiles. The lower and
upper bars are the minimum and maximum values (excluding an outlier for *L. laeve* under cell wall). The
solid middle line within the box is the median value and the dash middle line the average. The dashed
diamond box represents one standard deviation. The drawn-on cross represents the XRD mol% (X) and the
seasonal range (A) of mol% for the Arctic Bay – Kingitok – Quirpon dataset in figure 5.


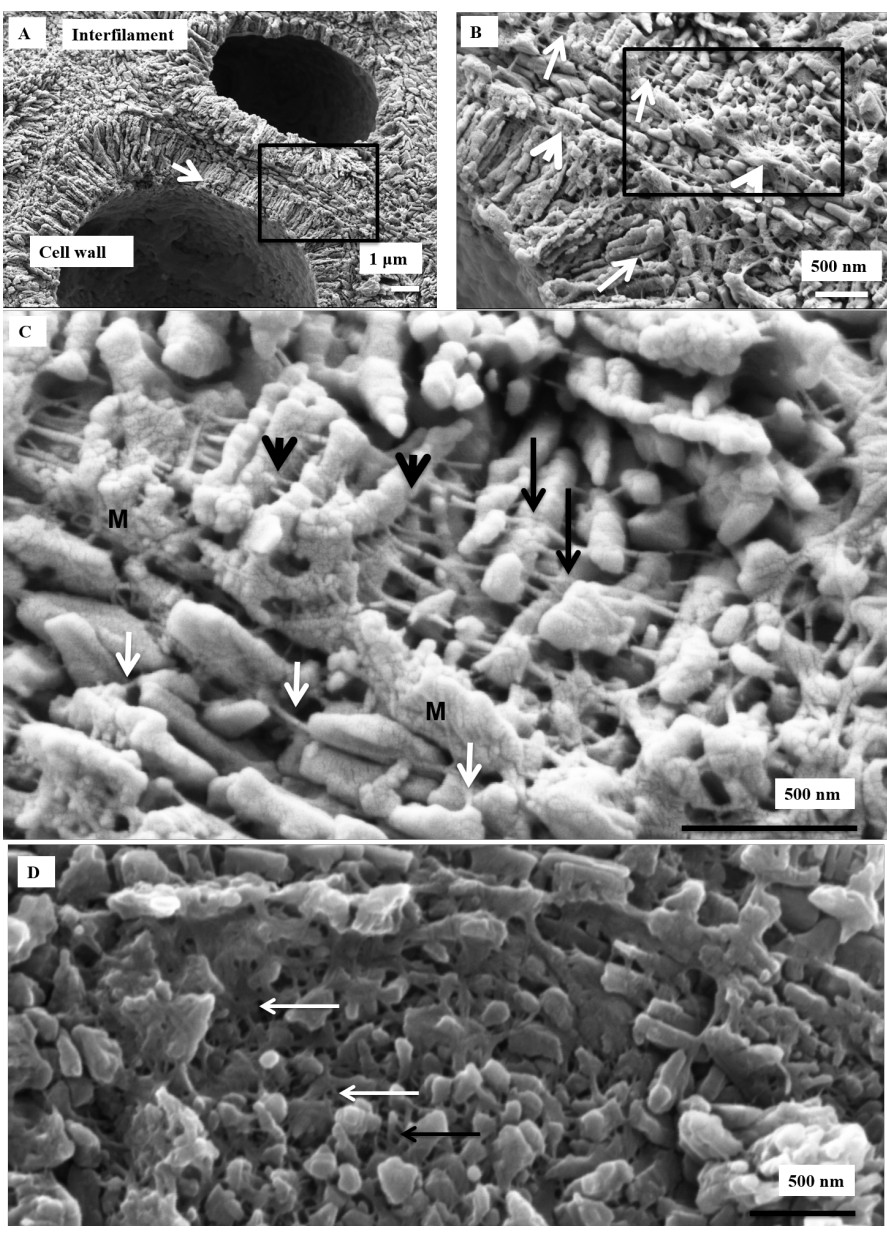


Figure 7: *K. epilaeve* cell wall structure. Crust polished and cleaned for 2 minutes. **A**. Cell walls have radial

Mg-calcite whereas the interfilament grains are orientated either parallel to the filament axis or randomly

within the corner junctions. Within the radial cell walls a secondary concentric banding pattern is visible

(white arrow). Black box enlarged in B. **B**. Organic fibrils, ~10nm wide, run parallel to cell wall edges



(black arrows).  Fibrils are concentrated along the outer of the cell wall (white arrows).  Black box enlarged
in C. **C**. The cell wall fibrils appear to string through the centre of the radial grains (black arrowheads),
Other fibrils drape over the grains (black arrows). Fibrils are present in the interfilament (white arrows). M
– mineralized membrane. **D**. Plan view of cell wall grains. Organic fibrils form a dense mesh (white
arrows).

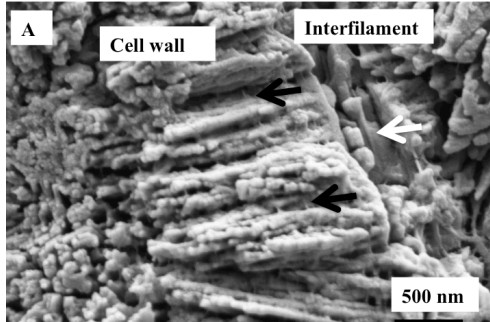 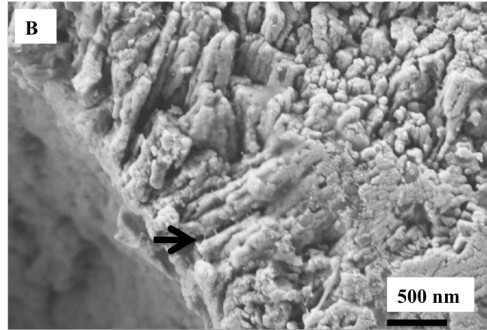


Figure 8: *L. laeve* cell wall structure. A. Cleaned for 2 minutes. Cell wall radial crystals are 1.5 micron
length cylindrical grains.  Fibrils are present (black arrows) but not as easy to see as in the *K. epilaeve*.
Interfilament grains parallel to cell wall with organic fibrils (white arrows) also running parallel to cell wall.
B. Etched for 20 minutes. Fibrils appear similarly as in the *K. epilaeve* with the fence post-wire structure
(black arrows).



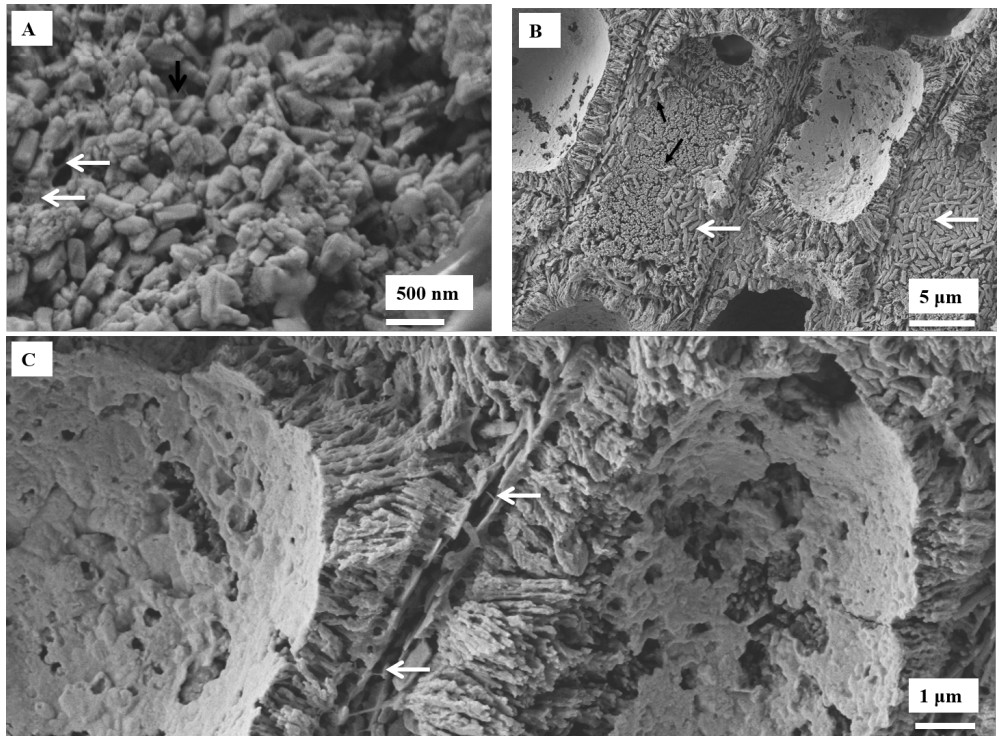


Figure 9: Interfilament structures in *K. epilaeve* (A) and *L. laeve* (B, C). **A**. *K. epilaeve* etched for 20
minutes. Fibrils (black arrow) and porous membrane (white arrows). **B**. *L. Laeve* etched for 20 minutes.
Interfilament grains are flattened against the external sides of the cell wall (white arrows) attached by
fibrils (black arrows). **C**. Fibrils visible stretched across the space between cell walls with 2 layers of
interfilament grains (white arrows).



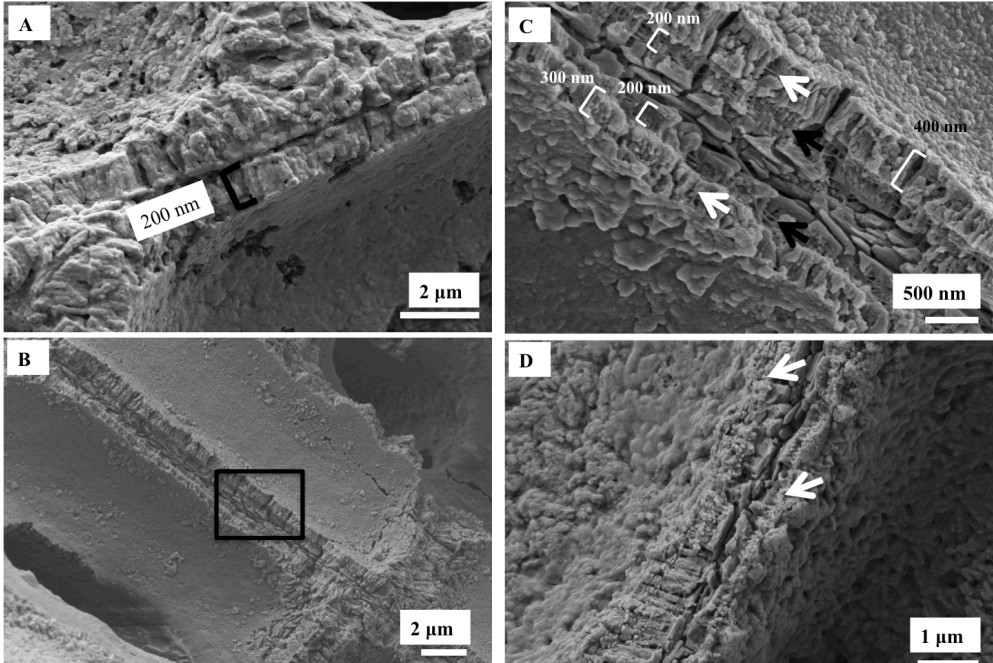


Figure 10: Hypothallus and transitional cells in *L. leave*. Cleaned 2 minutes. **A**. Hypothallus underside.
Organic film covering wall structures. Walls ~200 nm wide, roughly radial structure within cell wall. **B**.
Cleaned 2 minutes, hypothallus in upper crust. Roughly radial structure within cell walls. Black box
enlarged in C. **C**. The wall adjacent to the interfilament is narrowest at ~200 nm, has closely spaced organic
fibrils (black arrows) and is poorly calcified compared to the inner part of the wall (300-400 nm wide)
where radial grains are present. There are fibrils parallel to the cell wall appearing to go through the wall
grains similarly to the perithallial cell walls (white arrows). **D.** Transitional cell wall. The calcification in
the lower of the left side wall is comparable to the perithallial cell wall with radial grains. The right side
wall and upper part of the left side (white arrows) are poorly calcified and appear as a calcified membrane
rather than a properly developed cell wall.