# Peer review of "ANATOMICAL STRUCTURE OVERRIDES TEMPERATURE CONTROLS ON"

_Biogeosciences, 2017_

## Referee Comment (RC1) · J. Fietzke (Referee) · 19 Jun 2017

The submitted manuscript of Nash and Adey addresses important aspects of coralline algal skeletal features and their respective impact on using chemical proxies like Mg content for the reconstruction of environmental parameters i.e. ambient temperature.

This is significant, competently researched material. Obviously, I will not ridicule myself trying to lecture the authors on algal ecology and physiology or description of algal

skeletal features. In particular to my understanding the senior author is well-recognized as one of the world-experts in this field and a person I would ask for advice in such matters.

I do strongly belief the authors' approach is a very important one and we need more such studies to better understand the underlying processes controlling the formation of calcified hard-tissue of all kinds of biogenic samples on the $\mu$m and nm scale. This has the potential to greatly improve our conceptual understanding for the use of chemical proxies.

I've got no doubt this manuscript fits into the scope of BG and will become a valuable contribution. In its current form the manuscript strongly focusses on the investigation of skeletal ultrastructure and related Mg variability. That's in my opinion the real strength of this work. The consequences for the proxy application are discussed in the latest parts relatively briefly. I guess this study is supposed to be followed by more in the coming years and the mentioned discussion could be dealt with in a future manuscript in more detail. Nevertheless, I would underline the fact, that existing temperature-Mg calibrations used to be based on "bulk" methods, at least when compared to the ultrastructure studied in here. Applying a calibration derived from empirical correlation of temperature to mean concentrations (averaging tens or hundreds of $\mu$m) to sub-$\mu$m chemical variations may be critical, simply because it is an extrapolation beyond the factual base used to establish the former. Don't get me wrong, I consider this a valid point, just, please, be careful not to blame the existing calibrations not to work for the fine-scale variations. They never have been developed on that base and to my knowledge did not claim to work for anything beyond their spatial resolution. Thus, please, reconsider the point you make e.g. in figure 5 (how do the different skeletal parts contribute to the particular mix at a given time?).

I would love to see a more detailed quantitative evaluation of to what degree the seasonal variation in Mg content (for lower resolution studies) can be explained by changing skeletal structure, thus, just being an indirect response to temperature or even just

a co-variation with temperature due to seasonal changes in algal physiology. Or how much of the Mg-variation is truly coming from a changed chemical composition of the calcite crystals formed and how much is reflecting changing skeletal structure? But again, this may be the focus of future work and too early to address in this work.

As you cite Ragazzola et al. (2016) in the text, please, add this citation to the reference list.

Check wording in line 180 "show identify".

Finally, please, reconsider the very short conclusion. Is this all, you want to state?

Cheers,

jf

––––––––––––––––––––––––––

---

## Referee Comment (RC2) · A. Caragnano (Referee) · 5 Jul 2017

A. Caragnano (Referee)

annalisa.caragnano@unimib.it

The manuscript submitted by Nash and Adey and entitled "Anatomical structure overrides temperature controls on magnesium uptake – calcification in the Arctic/subarctic coralline algae Leptophytum laeve and Kvaleya epilaeve (Rhodophyta; Corallinales)" is an interesting study in biomineralization of coralline algae. The authors investigate important aspects of calcification in two genera of coralline algae (order Hapalidiales),

such as structural aspects of biomineralization, as well as the control by the plant vs. temperature in the magnesium uptake. I think that in the recent scenario of climate global change and of growing use of these organism as climate proxies we need more studies to improve our understanding of mechanism that lead mineralization in these plants. Nevertheless, I would underline some observations and suggestions arose during the careful reading of the manuscript.

The authors should enhance the literature cited. For example L75-78, are not there other published studies after the 1975 on coralline growth rates under different range of temperature and light conditions? Cabioch and Giraud (1986 In: Biomineralization in lower plants and animals [Ed. B.S.C. Leadbeater R. Reading], Clarendon Press, Oxford) reported a chapter entitled: Structural aspects of biomineralization in the coralline algae (calcified Rhodophyceae). The authors investigate the cytophysiological features of biomineralization in several examples of coralline algae from the different types of organization (crustose (M. lichenoides; L. lenormandii; L. sonderi; L. incrustans), branching and articulated (Jania rubens)). They found that calcification is a two-step phenomenon: 1) "In the outer zone the general envelope contains thin needles arranged tangential to the cells and parallel to the polysaccharide fibrils. Towards the base of the outer cells needles change progressively into plates. Among the epithallial cells, in the youngest parts, only tangential crystals can be observed and they are regularly arranged in the lateral walls." 2) "Inwards, from the perithallial meristem and directly under the epithallus, calcification increases and a second phase can be observed in the form of crystallization perpendicular to the cell wall. These secondary crystals are very closely juxtaposed and form in contact with the plasmalemma. After gentle decalcification they appear to be inserted between the radial polysaccharide fibrils. . . . . . .. The wall of each cell is made of a primary part, with tangential fibrils and crystallization, and of a secondary part with radial fibrils and crystallization. These observation show that biomineralization is controlled by the cells and that radial crystallization is a secondary process. . . . . . . . .".

The authors should replace the words epithallium, perithallium and hypotallium with epithallum, perithallum and hypotallum respectively. Indeed, the terms epithallus, perithallus and hypothallus are respectively formed with the prefix epi- (form the Greek áijŘ$\pi$Îŕ ‎= on top of), peri- (from the Greek $\pi\varepsilon$Îŕ ‎= around, close to), hypo- (from the Greek á¡Ś$\pi$Σ- = below) and the word thallus (from the Greek ÏŚ$\alpha\lambda\lambda$ÏŇς (offshoot, from thallein = to sprout), becoming thallum in Latin). Instead thallium is the chemical element.

Despite the interest in the topic of this study, one of my uncertainty concerns the resolution of EDS beam for measuring the Mg content in parts of the cell wall that range between $0.5\mu$m to $2\mu$m in thickness. The same authors raise the problem, and they write that the values measured for cell wall and interfilament may include small amount of the other. Moreover, although the authors could not know time and temperature of formation for each algal component (L403-404), they suggest that the magnesium offsets in different parts of the crust are clearly aligned to anatomical features and not controlled by temperature on base of a calibration value obtained on different species of CCA in experimental treatments (L410-417). I think that, although it could be possible, with these approximations the affirmation is not well supported.

L79: In the references there are two articles for Adey et al. 2015. Here, is it Adey et al 2015a?

L183: L. laeve (in italic)

L265-267: I think that for comparing the calcified cell structure of K. epilaeve with the one of L. laeve, the cells should be oriented in the same way. I suggest to embed the sample into epoxy resin for driving the cut.

L281-283: should it be moved in discussion.

L288: the authors reported that the lowest values of Mg content were for the K. epilaeve PIF and PCW. Could it due because the thallus was in a earlier stage of calcification

than the perithallial cells of L. laeve (Cabioch  Giraud 1986; see also L329-331)?

L347: remove space

Figures:

In some case, there is not conformity between the text and the figures:

L129: It should be Figs 1A and B

L134: It should be Figs 1C and D

L247: Fig. 2D?

L249: Fig. 3B?

L251-263: Should it be Fig. 3?

L269: in figure 3A is not possible to see the areas of interfilament.

L271: should it be Fig. 3B?

L272: should it be Fig. 4A, B?

L368: Maybe figure 1C?

L377: figure 1B do not show any layering

The figures should be cited in the text in order of the presentation, though figure 5 is taken from other articles (L211 the figure 5 is cited after figure 1 and before figure 2). I think that the authors should show the figures in the same order in the composed figures. For example in Fig. 1 the figures are ordered from left to right, on contrary in Fig. 2 they are ordered from the top to bellow.

All the Best,

AC

---

## Author Comment (AC1) · 13 Aug 2017

Nash and Adey
Response to reviewers

We thank both the reviewers for their positive comments and suggestions. We appreciate their recognition for the importance of this work.

The technical edits have been made, excluding the suggestion to reorder the mention of figure 5 in the MS. This is mentioned before figure 2 as it is relevant to the methods section, however it is logically placed in the main text as figure 5 after the information presented in figures 1-4.

The editorial suggestions have been incorporated or addressed as detailed below.

**Reviewer- A. Caragnano**

L75-78 *add extra references, including Cabioch and Giraud.* Extra references added. Additionally, Cabioch and Giraud also now referred to in the discussion.

*Suggestion to change epithallium to epithallum.*
While we appreciate the lesson in Greek and latin, the latter which I certainly could have benefited from during my formal education, using the spelling with ium is common in current phycology literature and we do not propose to use spelling contrary to that currently applied in published literature.

L265-267 *long cut-* noted. The perithallial cells subject for the main comparison were similarly orientated. The hypothallial cells here were of secondary interest as their Mg was not measured, nor was there a substantial bulk of the crust as hypothallial growth. No changes made.

Line 288- Cabioch and Giraud were referring to the difference between the epithallial cells (not analysed in this study) and the perithallial cells. Thus, this is unlikely to be the explanation for the measured difference in this paper. In the discussion the following has been added to discuss the reason for the lower Mg:

The *K. epilaeve* perithallial cells had lower Mg than the *L. leave* perithallial cells.

Cabioch and Giraud (1986) described the *perithallial* cells as being a later stage of

development than *epithallial* cells. Epithallial cells do not have fully developed rounded

cell walls of the perithallial cells (Adey 2015a, b). Although Mg-content of epithallial

carbonate is lower than the perithallial values (Diaz-Pulido et al. 2014, Nash et al. 2015,

2016), the lower Mg measured here is not considered a result of different cell type as the

*K. epilaeve* cell walls have the radial calcite similarly to the perithallial *L. Leave*, indicating that these are similarly well developed. Considering the time of collection in early summer, it is quite possible that the *K. epilaeve* growth closest to the *L. leave* surface was laid down closer to winter and in cooler temperatures, this being a likely explanation for the lower Mg content.

L404-
*The reviewer suggests the affirmation that the Mg change is not related to temperature is not well supported.*

We disagree. We have provide statistical support for our claim that the Mg is related to anatomy with the eds measurements for the cell wall v interfilament v hypothallial, all being statistically significantly different. In the MS we provide numerical comparisons for the difference in temperature that would be required to drive the Mg content change, temperature changes that are unrealistic for that geographical area. Since first writing this MS, further research we have undertaken has revealed the same Mg content patterns with anatomy for another general, *Phymatolithon*. This work has just been accepted for publication and we have added this reference (Nash and Adey 2017). We consider this extra data showing the same phenomenon supports our affirmation.

**Reviewer J. Fietzke**

*Regarding the reviewers concerns that we suggest existing calibrations not to work.*
This is not our intention, nor do we believe they do not work. The information in this MS will aid to refine future calibration. However, understanding that a switch to elongated thin-walled cells can involve an anatomically –driven increase in Mg has relevance for temperature calibrations made using rhodoliths that regularly switch to elongate, thin-walled higher Mg cells. We have made the following 2 edits:

Edit 1 line 506
We do not suggest current studies are inadequate because the finer scale (submicron) scale variations are not captured. These fine scale variations will not change the general trends or conclusions. Rather, we suggest caution regarding interpretation of data where a change in Mg is visibly associated with a change in cell type as temperature may not be the only possible driver of Mg change.

Edit 2 line 542

The rhodolith summer cells have similarities in appearance to the hypothallial cells in this study. Possibly the higher measured Mg in the long cells of the rhodolith is a result in part of a switch towards a more perithallial style cell and may not be entirely temperature related. This proposition is supported by Sletten et al. (2017) who found a switch to elongated cells with higher Mg that was unrelated to seasonality.

*More detailed quantitative evaluation of to what degree the seasonal variation in Mg content can be explained by changing skeletal structure.*
To do the seasonal quantitative evaluation we would need samples with the seasons constrained experimentally by stain or other geochemical tracer. It would be very interesting to do this work and would help to understand exactly how much the changing proportion can influence the total values. Unfortunately it is not something that is planned at this time.

*Or how much of the Mg variation is truly coming from a changed chemical composition of the calcite crystals formed and how much is reflecting changing skeletal structure.*
The reviewer is correct in that this question has been the focus of later work. Some of this work showing the consistent change with temperature for both interfilament and cell wall has just been accepted for publication and the reference has been added. We have added a brief mention in the concluding discussion on the extra work and edited the paragraph appropriately

Edit Line 568
 Recent work indicates that the interfilament and perithallial carbonate react similarly to temperature, but the responsive hypothallial carbonate is inconclusive (Nash and Adey 2017). It would be interesting to identify if each of interfilament, perithallial and hypothallial cell walls reacted similarly to changes in seawater Mg:Ca, or if there were differences in anatomical controls.

*Reconsider the very short conclusion*
We deliberately kept this short as it is the first of several papers investigating this topic and they each are getting longer and longer. We have however added a second short paragraph.

Edit

While the focus of this study has been the distribution of Mg with different anatomical

features, the high-magnification images are the first to show the cellular-scale organic

structures together with the carbonate components. The orientation of the crystals in the

interifilament and the cell walls are in agreement with lower-magnification SEM studies

on a range of algal species (Cabioch and Giraud 1986, Adey et al. 2013). The

combination of gentle etching and high-magnification SEM has revealed previously unknown features such as the fibrils threading through the radial Mg-calcite (Fig. 7C). Further, showing that the Mg content varies with anatomical features suggests that the calcification may be a different process, or have different controls, for each carbonate type.  This adds an extra level of complexity when considering how environmental changes, such as increasing temperature, may impact on the capacity of the CCA to continue their important substrate provision ecological role.

---

## Editor Comment (EC1) · L.J. de Nooijer (Editor) · 4 Sep 2017

Dear Mrs Nash and Mr Adey,

I have seen your replies to the reviewers, whose comments I think you have adequately addressed. Therefore, I think your manuscript is fit for publication in Biogeosciences. Upon reading your revised version, I came across a number of (minor) issues that I ask you to incorporate in your final version.

[Figure]

Sincerely,

Lennart de Nooijer

Abstract The last sentence is not relevant for this manuscript and I therefore suggest to remove it.

Introduction Lines 63-65: aren't there also studies reporting minor/ no effects of OA on coralline algal functioning? E.g. Cox et al., 2017. Mar Biol 164. Please include this nuance. Lines 87-90: I don't think it is desirable to use unpublished data as evidence for –in this case- continuous growth during day and night. Please include a reference to published results or remove this statement. Lines 97-102: this information belongs in the Methods section rather than the Introduction. Line 111: '(to one mm' should become '(up to one mm'.

Methods Line 131: unclear phrasing. Line 137: salinity is a unitless quantity. Line 148: I assume 'WHA' refers to one of the authors, but I don't see the relevance of mentioning this. Line 167: consider replacing 'enough' by 'sufficient' Lines 212-213: I think this can be removed here.

Results Line 218 and onwards: replace 'microns' by '$\mu$m'. Line 267: there should probably be an 'are' after 'these'

Discussion Line 457: elevated temperature in what? Do you mean differences in temperature through the year? Line 484-485: unclear phrasing. What is a 'magnesium perspective'? Line 527: do the authors mean that light could influence the Mg-content of the CCA? If so what exactly do they refer to? Light intensity? Day length? And are there references for such a suggestion? Line 550: should be 'ultrastructures'.

Figures Figure 2, 3 and 5 should have the species names italicized within the figures. Figure 2: 'L. laeve' is difficult to read. Consider changing the color to white or include a white background. Figure 2: the white arrows in panels B, D and E refer to a CaCO3 layer, rather than a single spot. Consider adding a line or band to indicate where these

layers begin and end. It is also not clear where the 'Meristem' in panel C refers to. Please also check for the other figures. Figure 3: the black circles in panel B are a bit difficult to see. Perhaps they are better visible as white circles. The caption does not mention what the white circles represent in panel C. For various names in the different panels, it is not clear to what part of the tissue they refer to exactly (see also previous comment). Figure 4: similar as to the comment to figure 3; are the black circles referring to the EDS sites? Figure 5: could you place the labels for the vertical axis next to the axis?

---

## Author Response (AR1)

Response to Editors comments

All edits have been incorporated.

Reference to meristem in figure to has been changed to epithallus and change has been made in the figure.
Figure 5- vertical axis line has been changed to black and thickened, labels are already on this axis.

[revised manuscript text omitted]